

# Analysis of the optimal patterns of serum alpha fetoprotein (AFP), AFP-L3% and protein induced by vitamin K absence or antagonist-II (PIVKA-II) detection in the diagnosis of liver cancers

Ling Luo*, Xiaochen Wang*, Xujian Peng, Runqiang Zhong, Xuejing Xuan, Haixiong Lin, Xianghua Lin and Chaohui Duan

Department of Clinical Laboratory, Sun Yat-Sen Memorial Hospital, Sun Yat-Sen University, Guangzhou, Guangdong, China
* These authors contributed equally to this work.

## ABSTRACT

**Background:** Liver cancers are common malignancies that primarily include hepatocellular carcinoma (HCC) and cholangiocarcinoma (CCA). Currently, the most commonly used serum markers for HCC are alpha fetoprotein (AFP), AFP-L3% and protein induced by vitamin K absence or antagonist-II (PIVKA-II), while the most commonly used serum markers for CCA are carcinoembryonic antigen (CEA) and carbohydrate antigen 19-9 (CA19-9). In recent years, many HCC diagnostic models using the combined detection of serum AFP, AFP-L3% and PIVKA-II have been established. For serum AFP, AFP-L3%, PIVKA-II and their many diagnostic models, there has been no clear guidance on the selection of these markers and their various combinations in the diagnosis of liver cancers. The aim of this study was to evaluate and compare the efficacy of these markers and the models that incorporate them in diagnosing HCC and CCA. This could assist in identifying the optimal patterns of serum AFP, AFP-L3% and PIVKA-II for the diagnosis of liver cancers.

**Methods:** Clinical data and the results of serum AFP, AFP-L3%, PIVKA-II, CEA and CA19-9 were collected from 117 patients with HCC, 28 patients with CCA and 101 patients with benign liver diseases. Laboratory tests and detection of serum tumor markers in liver cancer patients were conducted prior to treatments. Recently published diagnostic models for AFP, AFP-L3% and PIVKA-II detection were collected; these included GALAD, ASAP, GALAD-C, GAAP, C-GALAD, C-GALAD II and GAP-TALAD.

**Results:** Levels of AFP-L3%, PIVKA-II, GALAD, ASAP, GALAD-C, GAAP, C-GALAD and C-GALAD II significantly differed between the patient cohorts, with the highest levels seen in HCC, followed by CCA and with the lowest levels seen in benign liver diseases ($p < 0.05$). Levels of CEA and CA19-9 significantly differed between cohorts, with the highest levels seen in CCA, followed by HCC and with the lowest levels seen in benign liver diseases ($p < 0.05$). Levels of AFP and GAP-TALAD in HCC patients were significantly higher than those in patients with CCA and patients with benign liver diseases ($p < 0.05$), but there were no significant differences in levels of AFP and GAP-TALAD between patients with CCA and benign liver

Corresponding authors
Xianghua Lin,
lxiangh@mail.sysu.edu.cn
Chaohui Duan,
duanchh@mail.sysu.edu.cn

diseases ($p > 0.05$). In the diagnosis of HCC, GAP-TALAD, GALAD, C-GALAD, ASAP and GALAD-C showed the highest efficacy. In the diagnosis of overall liver cancers (HCC and CCA), GALAD-C, GAAP, GALAD, ASAP and C-GALAD showed the highest efficacy. In the diagnosis of early liver cancers (early HCC and CCA), GALAD, GALAD-C, GAAP, C-GALAD and ASAP showed the highest efficacy.

**Conclusions:** For serum AFP, AFP-L3% and PIVKA-II, diagnostic models of combined marker detection improved efficacy in the diagnosis of liver cancers. Diagnostic models GALAD, ASAP, GALAD-C and C-GALAD showed the highest efficacy in the diagnosis of HCC, overall liver cancers (HCC + CCA) and early liver cancers, and can be used preferentially in clinical practice.

# INTRODUCTION

Liver cancers are common malignancies that pose a serious health concern. Worldwide, hepatocellular carcinoma (HCC) is the most prevalent type of liver cancer (about 75–85% of cases), the sixth most common malignancy type and the fourth leading cause of cancer-related death (*Brown et al., 2023*; *Frager & Schwartz, 2020*; *Sung et al., 2021*). Cholangiocarcinoma (CCA) comprises about 15% of liver cancer cases, and the worldwide incidence and mortality rates of CCA are consistently increasing (*Sarcognato et al., 2021*). The high mortality rate of liver cancers is partly due to the lack of timely and accurate diagnosis; many patients are diagnosed when the disease is already at an advanced stage (*de Martel et al., 2020*). The detection of tumor markers in blood is a simple, convenient and important method in the diagnosis of liver cancers. As such, studying blood tumor markers for liver cancers is a popular field in current medical research.

Alpha fetoprotein (AFP) is a type of glycoprotein that is synthesized by the yolk sac during early foetal life and later by the foetal liver (*Debruyne & Delanghe, 2008*). Under normal circumstances, hepatocytes in adults do not synthesize AFP; however, during the malignant transformation of hepatocytes, the AFP gene is activated and serum AFP level increases (*Debruyne & Delanghe, 2008*; *Huang et al., 2023*). AFP can be divided into three glycoforms based on how strongly they bind to lens culinaris agglutinin (LCA): AFP-L1 (nonbinding), AFP-L2 (weak binding) and AFP-L3 (strong binding). AFP-L3 is produced by cancerous hepatocytes, and the percentage of AFP-L3 in total AFP is called AFP-L3% (*Dunbar, Kushnir & Yang, 2023*; *Hanif et al., 2022*). Protein induced by vitamin K absence or antagonist-II (PIVKA-II), also known as des-gamma carboxyprothrombin (DCP), is considered a defective prothrombin protein and is produced from prothrombin precursor when the ten glutamate residues in the N-terminal of prothrombin precursor cannot be fully carbonylated (*Dong et al., 2023*). The carboxylation of prothrombin precursor in the liver can be dysfunctional in HCC, and this leads to the increased production of PIVKA-II (*Card, Gorska & Harrington, 2020*; *Dong et al., 2023*; *Svobodova et al., 2018*). Serum AFP,

AFP-L3% and PIVKA-II are the most commonly used blood markers for HCC (*Huang et al., 2023*; *Liu et al., 2022*), and the most commonly used serum markers for CCA are carcinoembryonic antigen (CEA) and carbohydrate antigen 19-9 (CA19-9; *Shin, Moon & Kim, 2023*).

The diagnostic efficacy of a single detection of tumor markers for HCC is limited; serum AFP, AFP-L3% and PIVKA-II still lack high sensitivity and specificity in the diagnosis of HCC (*Dong et al., 2023*; *Hanif et al., 2022*). Many diagnostic models for the combined detection of AFP, AFP-L3% and PIVKA-II have been proposed in recent years. These models use the combined detection of serum tumor markers (AFP, AFP-L3% and PIVKA-II) alongside demographic information, laboratory test results and specific algorithms, including GALAD (gender, age, AFP, AFP-L3% and PIVKA-II) and GALAD-like models (*Huang et al., 2023*). Serum AFP, AFP-L3%, PIVKA-II and the diagnostic models based on these markers provide more methods and options for the diagnosis of HCC, but they also present some limitations. There has been no comparison of diagnostic efficacy between serum tumor markers and different diagnostic models, complicating the selection of blood tumor markers and/or diagnostic models in clinical practice. Additionally, the diagnostic models (GALAD model, GALAD-like models, *etc.*) were all established from HCC patients; these measurements in CCA patients and the differences between HCC and CCA patients are also unclear.

This study analyzed and compared diagnostic efficacy for liver cancers (including HCC and CCA) in serum AFP, AFP-L3%, PIVKA-II and recently published diagnostic models containing combined detection of these markers, as well as in serum CEA and CA19-9. The diagnostic models included GALAD (*Johnson et al., 2014*), ASAP (*Yang et al., 2019*), GALAD-C (*Liu et al., 2020*), GAAP (*Liu et al., 2020*), C-GALAD (*Qi & Lin, 2021*), C-GALAD II (*Li et al., 2022*) and GAP-TALAD (*Li et al., 2022*). The aim of this study was to find the optimal patterns of serum AFP, AFP-L3% and PIVKA-II for the diagnosis of liver cancers to provide clearer guidance on the selection of hematological diagnostic methods for liver cancers in clinical practice.

## MATERIALS AND METHODS

### Study subjects and design

This study retrospectively collected clinical data and results of serum tumor markers for liver cancers (AFP, AFP-L3%, PIVKA-II, CEA and CA19-9) from 117 patients with HCC, 28 patients with CCA and 101 patients with benign liver diseases (including 31 cases of hepatic cyst, 23 cases of hepatic hemangioma, 22 cases of liver cirrhosis, nine cases of hepatic abscess, seven cases of focal nodular hyperplasia, three cases of chronic hepatitis B, two cases of fatty liver disease, two cases of angiomyolipoma, one case of hepatic adenoma and one case of falciform ligament pseudolesion) at the Sun Yat-Sen Memorial Hospital, Sun Yat-Sen University from February 2023 to January 2024. HCC and CCA were diagnosed in accordance with clinical guidelines (*Zhang et al., 2022*; *Zhou et al., 2023*), and all laboratory tests and serum tumor marker analyses in the study were conducted before treatments. Patients were excluded from the study based on the following exclusion criteria: (1) patients with malignancies other than HCC and CCA, and with

choledocholithiasis and acute cholangitis; (2) patients with recurrent HCC or CCA; (3) patients with hemorrhagic or thrombotic diseases and those who had previously taken anti-coagulant agents or vitamin K; and (4) patients who were pregnant. The results of the diagnostic models were calculated according to the formulas in Table 1. Baseline clinical data, levels of serum tumor markers and diagnostic models in all groups are listed in Table 2. This study was approved by the Medical Ethics Committee of Sun Yat-Sen Memorial Hospital, Sun Yat-Sen University, Guangzhou, China (SYSKY-2024-317-01). Informed consent was not required because this study was observational and did not involve the personal health information of patients.

This study first analyzed the levels of serum AFP, AFP-L3%, PIVKA-II, CEA, CA19-9 and results from diagnostic models (GALAD, ASAP, GALAD-C, GAAP, C-GALAD, C-GALAD II and GAP-TALAD) in HCC, CCA and benign liver diseases. Then, the diagnostic efficacies of the markers and models for HCC, overall liver cancers (HCC and CCA) and early liver cancers were evaluated and compared to find the optimal patterns of serum AFP, AFP-L3% and PIVKA-II for the diagnosis of liver cancers.

## Laboratory tests and detection of serum tumor markers

Fasting venous blood samples collected at early morning in non-anticoagulant tubes were centrifuged at room temperature at 3,000 rpm for 10 min after at least 30 min standing to obtain serum samples. Blood platelet counts (PLT) were measured by electrical impedance in a Sysmex XN-9000 automated hematology analyzer (Sysmex Corporation, Kobe, Japan) using blood samples collected at the same time in EDTA-$K_2$ anticoagulant tubes. Serum total bilirubin (TBIL) and albumin (ALB) levels were measured in a Roche cobas c702 automatic biochemical analyzer using Roche BILT3 Kits and ALB2 Kits, respectively (Roche Diagnostics GmbH, Mannheim, Germany). TBIL was measured using the diazonium method, and ALB was measured using the bromocresol green method. Serum AFP, CEA and CA19-9 levels were measured by electrochemiluminescence assays using a Roche cobas e801 immunoassay analyzer and Roche Elecsys AFP, CEA and CA19-9 Kits (Roche Diagnostics GmbH, Mannheim, Germany). Serum AFP-L3% levels were measured by chemiluminescence immunoassays using a Hotgen C2000 immunoassay analyzer and Hotgen AFP-L3% Kits (Hotgen Biotech Co., Ltd., Beijing, China). Serum PIVKA-II levels were measured by chemiluminescence immunoassays using an ARCHITECT i2000SR immunoassay analyzer and PIVKA-II Reagent Kits (Abbott GmbH, Wiesbaden, Germany).

## Statistical analysis

Continuous variables with normal distributions were expressed as the mean ± standard deviation, and comparisons between two groups were performed by t-tests. Continuous variables with nonnormal distributions were expressed as the median (interquartile range), and comparisons between two groups were performed by Mann-Whitney U-tests. Chi-square ($\chi^2$) tests were used for the comparisons of categorical variables. Receiver operating characteristic (ROC) curves and the area under ROC curve (AUC) were used to evaluate diagnostic efficacy, and AUC was compared by DeLong tests. When the results of

**Table 1  Formulas of diagnostic models.**

| Diagnostic model | Formula | Reference |
|---|---|---|
| GALAD | $-10.08 + 1.67 \times$ [Gender (1 for male, 0 for female)] $+ 0.09 \times$ [Age] $+ 0.04 \times$ [AFP-L3%] $+ 2.34 \times \log_{10}$[AFP] $+ 1.33 \times \log_{10}$[PIVKA-II] | *Johnson et al. (2014)* |
| ASAP | $-7.5771177 + 0.04666357 \times$ [Age] $- 0.57611693 \times$ [Gender (0 for male, 1 for female)] $+ 0.42243533 \times \ln$[AFP] $+ 1.1051891 \times \ln$[PIVKA-II] | *Yang et al. (2019)* |
| GALAD-C | $-11.501 + 0.733 \times$ [Gender (1 for male, 0 for female)] $+ 0.099 \times$ [Age] $+ 0.073 \times$ [AFP-L3%] $+ 0.84 \times \log_{10}$[AFP] $+ 2.364 \times \log_{10}$[PIVKA-II] | *Liu et al. (2020)* |
| GAAP | $-11.203 + 0.699 \times$ [Gender (1 for male, 0 for female)] $+ 0.094 \times$ [Age] $+ 1.076 \times \log_{10}$[AFP] $+ 2.376 \times \log_{10}$[PIVKA-II] | *Liu et al. (2020)* |
| C-GALAD | $-8.654 + 0.044 \times$ [Age] $+ 1.329 \times$ [Gender (1 for male, 0 for female)] $+ 0.063 \times$ [AFP-L3%] $+ 0.885 \times \log_{10}$[AFP] $+ 3.138 \times \log_{10}$[PIVKA-II] | *Qi & Lin (2021)* |
| C-GALAD II | $-8.1942 + 0.114 \times$ [Age] $+ 0.8829 \times$ [Gender (1 for male, 0 for female)] $+ 0.761 \times \log_{10}$[AFP] $+ 0.0965 \times$ [AFP-L3%] $+ 1.0855 \times \log_{10}$[PIVKA-II] $- 0.0181 \times$ [PLT] $- 0.0043 \times$ [TBIL] | *Li et al. (2022)* |
| GAP-TALAD | $-16.01 + 0.064 \times$ [Age] $+ 1.569 \times$ [Gender (1 for male, 0 for female)] $- 0.005 \times$ [PLT] $- 0.06 \times$ [TBIL] $+ 0.161 \times$ [ALB] $+ 0.077 \times$ [AFP-L3%] $+ 1.38 \times \log_{10}$[AFP] $+ 3.858 \times \log_{10}$ [PIVKA-II] | *Li et al. (2022)* |

**Note:**
AFP, alpha fetoprotein; AFP-L3%, percentage of AFP-L3 (culinaris agglutinin strong binding) to total AFP; PIVKA-II, protein induced by vitamin K absence or antagonist-II; PLT, blood platelet counts; TBIL, total bilirubin; ALB, albumin.

**Table 2  Clinical data, levels of liver cancer serum tumor markers and diagnostic models in all groups.**

| | HCC ($n = 117$) | CCA ($n = 28$) | Benign liver diseases ($n = 101$) |
|---|---|---|---|
| Age (years) | $56.85 \pm 10.99$[a] | $61.64 \pm 10.05$[b] | $50.47 \pm 15.09$ |
| Gender (male%) | 86.32% (101/117)[a,c] | 50.00% (14/28) | 44.55% (45/101) |
| TBIL (μmol/L) | 16.80 (12.25, 27.60)[a] | 16.95 (11.65, 204.10) | 14.30 (11.20, 19.05) |
| PLT ($\times 10^9$/L) | 194.00 (132.00, 249.00) | 223.00 (170.80, 307.50) | 216.00 (166.00, 282.00) |
| ALB (g/L) | 36.60 (32.50, 39.85) | 35.50 (33.48, 39.90) | 38.50 (33.00, 41.70) |
| AFP-L3% | 10.99 (5.00, 17.55)[a,c] | 5.00 (5.00, 5.00)[b] | 5.00 (5.00, 5.00) |
| PIVKA-II (mAU/mL) | 747.30 (69.27, 12,807.00)[a,c] | 30.42 (20.91, 165.40)[b] | 26.28 (19.23, 33.86) |
| GALAD | 5.54 (2.53, 10.29)[a,c] | 0.61 (−1.01, 2.17)[b] | −1.45 (−2.69, −0.41) |
| ASAP | 4.45 (0.86, 8.39)[a,c] | 0.14 (−1.05, 1.47)[b] | −1.40 (−1.96, −0.81) |
| GALAD-C | 4.64 (1.12, 8.01)[a,c] | 0.34 (−0.98, 1.42)[b] | −1.82 (−2.93, −0.76) |
| GAAP | 4.15 (0.53, 7.68)[a,c] | 0.06 (−1.24, 1.16)[b] | −2.08 (−3.08, −0.98) |
| C-GALAD | 7.29 (2.22, 12.04)[a,c] | 1.11 (−0.42, 2.60)[b] | −0.66 (−1.35, 0.32) |
| C-GALAD II | 1.54 (−0.58, 3.55)[a,c] | −2.04 (−3.84, −0.32)[b] | −3.44 (−5.61, −1.73) |
| AFP (ng/mL) | 145.00 (6.63, 4,464.00)[a,c] | 2.96 (1.88, 8.14) | 2.84 (1.87, 5.13) |
| GAP–TALAD | 7.77 (2.35, 13.25)[a,c] | −1.72 (−7.98, 0.89) | −1.82 (−3.05, −0.58) |
| CEA (ng/mL) | 2.70 (1.70, 3.85)[a,c] | 4.15 (2.16, 9.75)[b] | 1.70 (0.80, 2.70) |
| CA19-9 (U/mL) | 24.00 (8.85, 38.90)[a,c] | 142.50 (44.75, 927.80)[b] | 9.30 (5.85, 23.35) |

**Notes:**
[a] HCC *vs.* benign liver diseases, $p < 0.05$.
[b] CCA *vs.* benign liver diseases, $p < 0.05$.
[c] HCC *vs.* CCA, $p < 0.05$.
HCC, hepatocellular carcinoma; CCA, cholangiocarcinoma; TBIL, total bilirubin; PLT, blood platelet count; ALB, albumin; AFP, alpha fetoprotein; AFP-L3%, percentage of AFP-L3 (culinaris agglutinin strong binding) to total AFP; PIVKA-II, protein induced by vitamin K absence or antagonist-II; CEA, carcinoembryonic antigen; CA19-9, carbohydrate antigen 19-9.

laboratory tests and serum tumor markers were lower or higher than the corresponding clinical reportable range, the value of the lower or upper limit of the reportable range was used in analyses. Statistical analysis was performed by SPSS 23.0, GraphPad Prism 9.0.0 and MedCalc v22.009 software, and $p$-values less than 0.05 were considered statistically significant.

## RESULTS

### Clinical data, levels of liver cancer serum tumor markers and diagnostic models in all study groups

The average ages of HCC and CCA patients were significantly higher than those of patients with benign liver diseases ($p < 0.05$), and there was no significant difference in age between HCC and CCA patients ($p > 0.05$; Table 2, Fig. 1A). The proportion of males in the HCC patient cohort was significantly higher than that in the CCA and benign liver disease cohorts ($p < 0.0001$), and there was no significant difference in gender composition between CCA patients and patients with benign liver diseases ($p = 0.6090$; Table 2). Median serum TBIL level in CCA patients was not significantly different from that in HCC patients and patients with benign liver diseases ($p > 0.05$), and median serum TBIL level in HCC patients was significantly higher than that in patients with benign liver diseases ($p < 0.05$; Table 2, Fig. 1B). There were no significant differences in PLT and serum ALB among patients with HCC, CCA and benign liver diseases ($p > 0.05$; Table 2, Figs. 1C and 1D).

Levels of AFP-L3%, PIVKA-II, GALAD, ASAP, GALAD-C, GAAP, C-GALAD and C-GALAD II significantly differed between the patient cohorts, with the highest levels seen in HCC, followed by CCA and with the lowest levels seen in benign liver diseases ($p < 0.05$; Table 2, Figs. 2A–2H). Levels of AFP and GAP-TALAD in HCC patients were significantly higher than those in CCA patients and patients with benign liver diseases ($p < 0.05$), and there were no significant differences in levels of AFP and GAP-TALAD between CCA patients and patients with benign liver diseases ($p > 0.05$; Table 2, Figs. 2I and 2J). Levels of CEA and CA19-9 significantly differed between cohorts, with the highest levels seen in CCA, followed by HCC and with the lowest levels seen in benign liver diseases ($p < 0.05$; Table 2, Figs. 2K and 2L).

### Efficacy of serum tumor markers and diagnostic models in diagnosing HCC

Levels of AFP-L3%, PIVKA-II, GALAD, ASAP, GALAD-C, GAAP, C-GALAD, C-GALAD II, AFP and GAP-TALAD in HCC patients were significantly ($p < 0.05$) higher than those in non-HCC (CCA and benign liver diseases) patients (Table 2). In the diagnosis of HCC and non-HCC (CCA and benign liver diseases), GAP-TALAD, GALAD, C-GALAD, ASAP and GALAD-C models showed the highest efficacy (AUC), and there was no significant difference ($p > 0.05$) in AUC among GAP-TALAD, GALAD, C-GALAD, ASAP and GALAD-C (Table 3 and S1, Fig. 3A).

Levels of serum CEA and CA19-9 in CCA patients were significantly ($p < 0.05$) higher than those in HCC patients (Table 2). There was no significant difference in efficacy

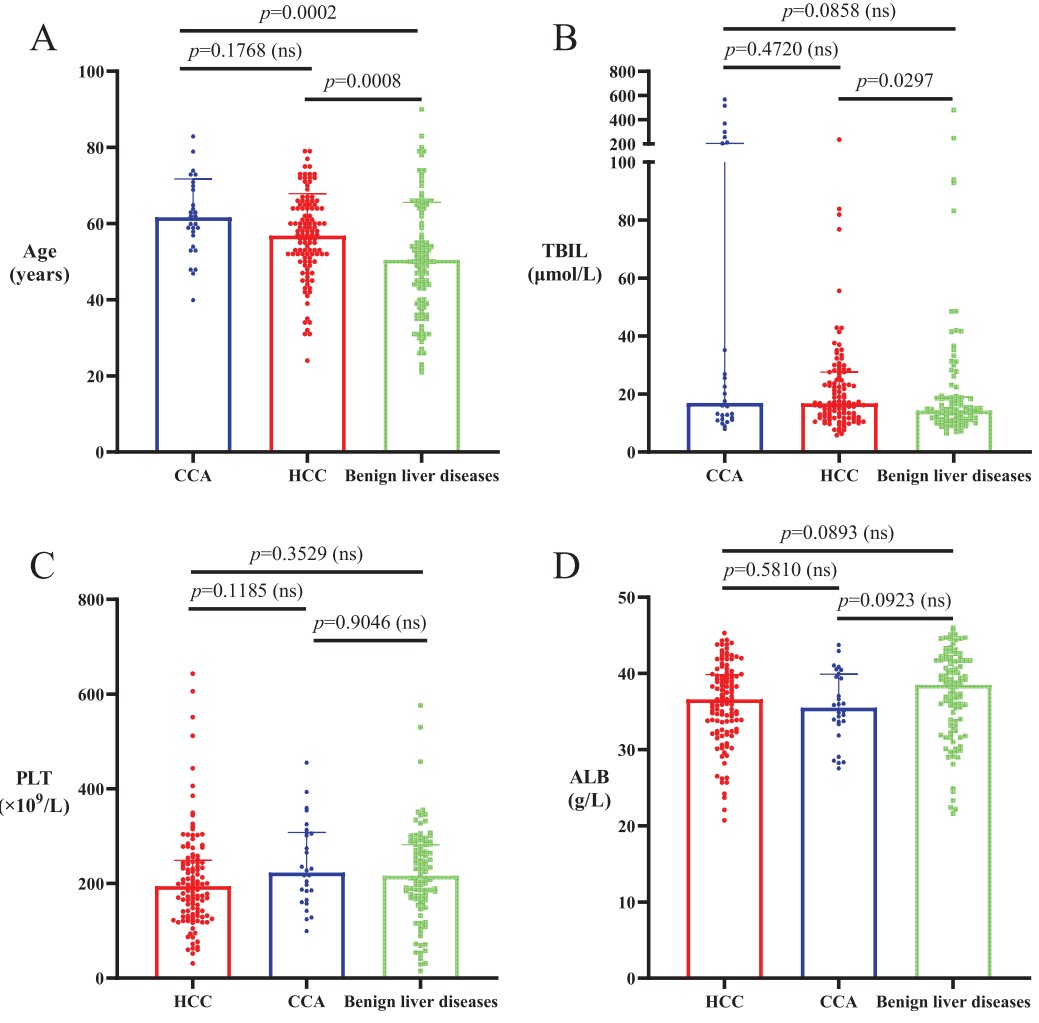

**Figure 1 Age and levels of TBIL, PLT and ALB in HCC, CCA and benign liver diseases patient cohorts.** (A) The average ages of HCC and CCA patients were significantly higher than the ages of patients with benign liver diseases ($p < 0.05$). There was no significant difference in average age between HCC and CCA patients ($p > 0.05$). (B) Serum TBIL level in CCA patients was not significantly different from that in HCC patients and patients with benign liver diseases ($p > 0.05$), and serum TBIL level in HCC patients was significantly higher than that in patients with benign liver diseases ($p < 0.05$). (C and D) There was no significant difference in PLT and serum ALB among patients with HCC, CCA or benign liver diseases ($p > 0.05$).

(AUC) of CEA and CA19-9 in distinguishing HCC and CCA ($p = 0.0579$). The AUC of serum CA19-9 and CEA in distinguishing HCC and CCA was also lower than the AUC of AFP, AFP-L3%, PIVKA-II and the diagnostic models based on these markers in diagnosing HCC and non-HCC (Table 3, Fig. 3B).

## Efficacy of serum tumor markers and diagnostic models in diagnosing liver cancers

Levels of AFP-L3%, PIVKA-II, GALAD, ASAP, GALAD-C, GAAP, C-GALAD, C-GALAD II, CEA and CA19-9 in liver cancer (HCC and CCA) patients were significantly

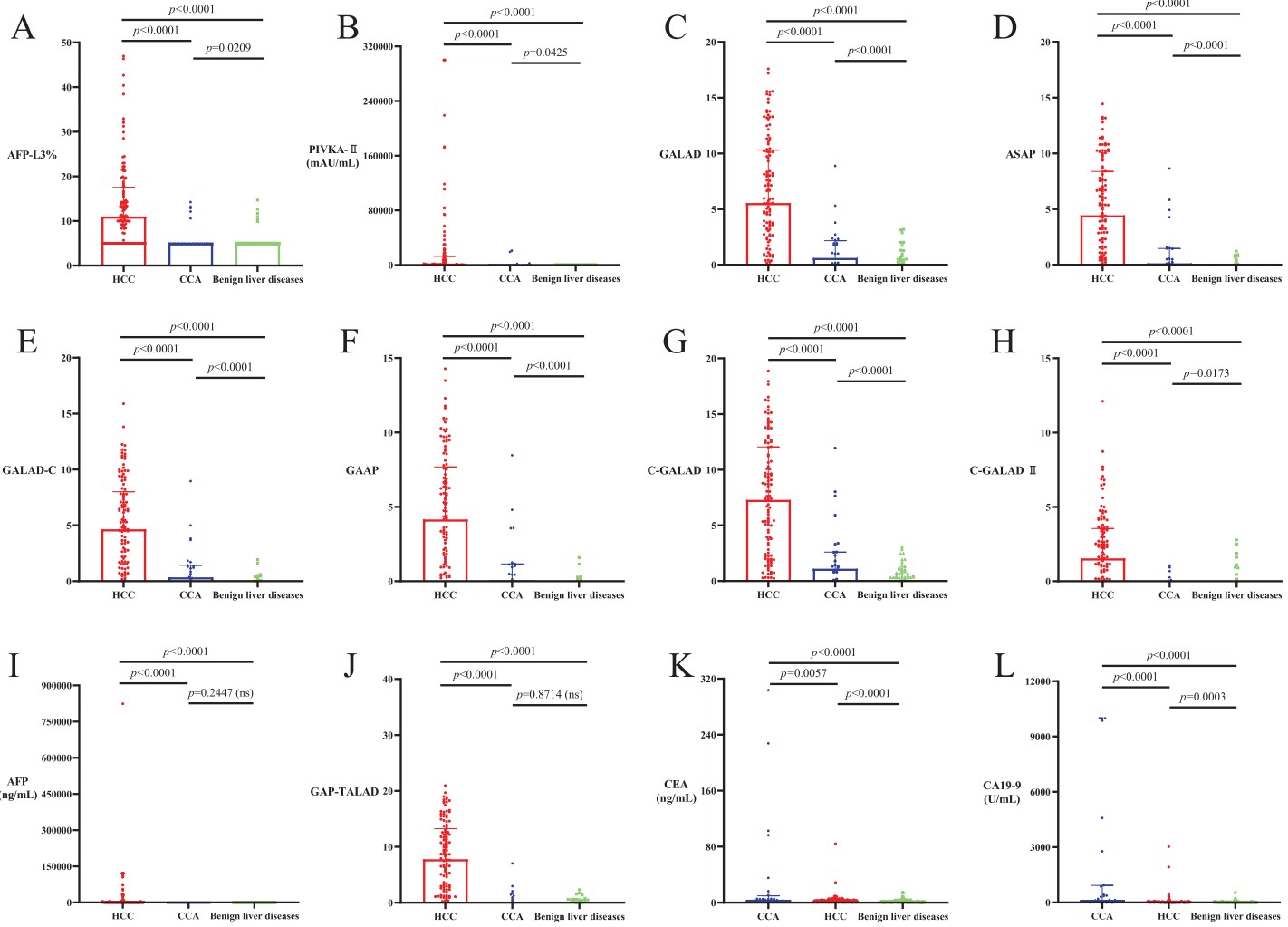

**Figure 2 Levels of liver cancer serum tumor markers and diagnostic models in HCC, CCA and benign liver diseases patient cohorts.** (A–H) Levels of AFP-L3%, PIVKA-II, GALAD, ASAP, GALAD-C, GAAP, C-GALAD and C-GALAD II significantly differed between the patient cohorts, with the highest levels seen in HCC, followed by CCA and with the lowest levels seen in benign liver diseases ($p < 0.05$). (I and J) Levels of AFP and GAP-TALAD in HCC patients were significantly higher than those in patients with CCA or benign liver diseases ($p < 0.05$), and there was no significant difference in levels of AFP and GAP-TALAD between CCA patients and patients with benign liver diseases ($p > 0.05$). (K and L) Levels of CEA and CA19-9 significantly differed between cohorts, with the highest levels seen in CCA, followed by HCC and with the lowest levels seen in benign liver diseases ($p < 0.05$).

($p < 0.05$) higher than those in patients with benign liver diseases (Table 2). In the diagnosis of liver cancers (HCC + CCA) and benign liver diseases, GALAD-C, GAAP, GALAD, ASAP and C-GALAD models showed the highest diagnostic efficacy (AUC), and there was no significant difference ($p > 0.05$) in AUC among GALAD-C, GAAP, GALAD, ASAP and C-GALAD (Tables 4 and S2, Fig. 4).

## Efficacy of serum tumor markers and diagnostic models in diagnosing early liver cancers

HCC cases with Barcelona clinic liver cancer (BCLC) stages 0 to A and CCA cases with TNM (primary tumor, regional lymph nodes and distant metastasis) stages 0 to II are

**Table 3 Efficacy of serum tumor markers and diagnostic models in diagnosing HCC.**

| Diagnosing/distinguishing | Serum tumor marker/model | AUC (95% CI)[#] | Cut-off value | Sensitivity% | Specificity% | Youden index |
|---|---|---|---|---|---|---|
| HCC and non-HCC[◆] | GAP-TALAD | 0.9445 [0.9148~0.9742][a,b] | 0.68[*] | 88.89% | 90.70% | 0.7959 |
| | GALAD | 0.9288 [0.8981~0.9595][a,c] | 2.03[*] | 78.63% | 92.25% | 0.7088 |
| | C-GALAD | 0.9227 [0.8900~0.9555][a,d] | 1.35[*] | 85.47% | 85.27% | 0.7074 |
| | ASAP | 0.9190 [0.8849~0.9530][a,d] | 0.33[*] | 82.05% | 86.82% | 0.6887 |
| | GALAD-C | 0.9174 [0.8837~0.9511][a,d] | 0.45[*] | 82.05% | 86.82% | 0.6887 |
| | GAAP | 0.9155 [0.8813~0.9496][d] | 0.19[*] | 82.05% | 86.82% | 0.6887 |
| | C-GALAD II | 0.8890 [0.8473~0.9306][e] | −1.77[*] | 91.45% | 72.87% | 0.6432 |
| | PIVKA-II | 0.8537 [0.8034~0.9040][f] | 40.00 mAU/mL[*] | 80.34% | 80.62% | 0.6096 |
| | AFP | 0.8535 [0.8031~0.9039][f] | 7.00 ng/mL[*] | 74.36% | 86.05% | 0.6041 |
| | AFP-L3% | 0.8197 [0.7643~0.8752][f] | 10.00%[*] | 52.99% | 92.25% | 0.4524 |
| CCA and HCC | CA19-9 | 0.8007 [0.6929~0.9085][g] | 34.00 U/mL[*] | 78.57% | 68.38% | 0.4695 |
| | CEA | 0.6670 [0.5415~0.7925][g] | 5.00 ng/mL[*] | 46.43% | 88.03% | 0.3446 |

**Notes:**
[#] Comparisons of AUC were conducted by DeLong tests.
[◆] CCA and benign liver diseases.
[*] According to the highest Youden index.
[*] According to the manufacturers' instructions.
[a] There was no significant difference ($p > 0.05$) in AUC (diagnosing HCC and non-HCC) among GAP-TALAD, GALAD, C-GALAD, ASAP and GALAD-C.
[b] AUC (diagnosing HCC and non-HCC) of GAP-TALAD was significantly higher ($p < 0.05$) than that of GAAP, C-GALAD II, PIVKA-II, AFP and AFP-L3%.
[c] AUC (diagnosing HCC and non-HCC) of GALAD was significantly higher ($p < 0.05$) than that of C-GALAD II, PIVKA-II, AFP and AFP-L3%.
[d] AUC (diagnosing HCC and non-HCC) of C-GALAD, ASAP, GALAD-C and GAAP was significantly higher ($p < 0.05$) than that of PIVKA-II, AFP and AFP-L3%.
[e] AUC (diagnosing HCC and non-HCC) of C-GALAD II was significantly higher ($p < 0.05$) than that of AFP-L3%.
[f] There was no significant difference ($p > 0.05$) in AUC (diagnosing HCC and non-HCC) among PIVKA-II, AFP and AFP-L3%.
[g] There was no significant difference ($p > 0.05$) in AUC (distinguishing CCA and HCC) between CA19-9 and CEA.
HCC, hepatocellular carcinoma; AUC, area under receiver operating characteristic (ROC) curve; CI, confidence interval; PIVKA-II, protein induced by vitamin K absence or antagonist-II; AFP, alpha fetoprotein; AFP-L3%, percentage of AFP-L3 (culinaris agglutinin strong binding) to total AFP; CCA, cholangiocarcinoma; CA19-9, carbohydrate antigen 19-9; CEA, carcinoembryonic antigen.

usually considered to be early stage cancers (*European Association for the Study of the Liver, 2018*; *Khuntikeo et al., 2020*). Within the serum tumor markers and diagnostic models for liver cancers (Table 4), GALAD, GALAD-C, GAAP, C-GALAD and ASAP models showed the highest diagnostic efficacy (AUC) in distinguishing between early liver cancers (including early HCC and CCA) and benign liver diseases (Table 5). There was no significant difference ($p > 0.05$) in AUC among GALAD, GALAD-C, GAAP, C-GALAD and ASAP (Tables 5 and S3, Fig. 5A).

Levels of GAT-TALAD and AFP showed no significant difference between CCA patients and patients with benign liver diseases (Table 2), and the efficacy (AUC) of GAP-TALAD in diagnosing early HCC and non-HCC (CCA and benign liver diseases) was significantly higher than that of AFP ($p = 0.0032$; Table 5, Fig. 5B). Additionally, the AUC of GAP-TALAD in diagnosing early HCC was higher than the AUC of PIVKA-II, AFP-L3%, CA19-9, CEA and other diagnostic models in diagnosing early liver cancers; however, the AUC of AFP in diagnosing early HCC was also lower than the AUC of GALAD, GALAD-C, GAAP, C-GALAD and ASAP in diagnosing early liver cancers (Table 5).

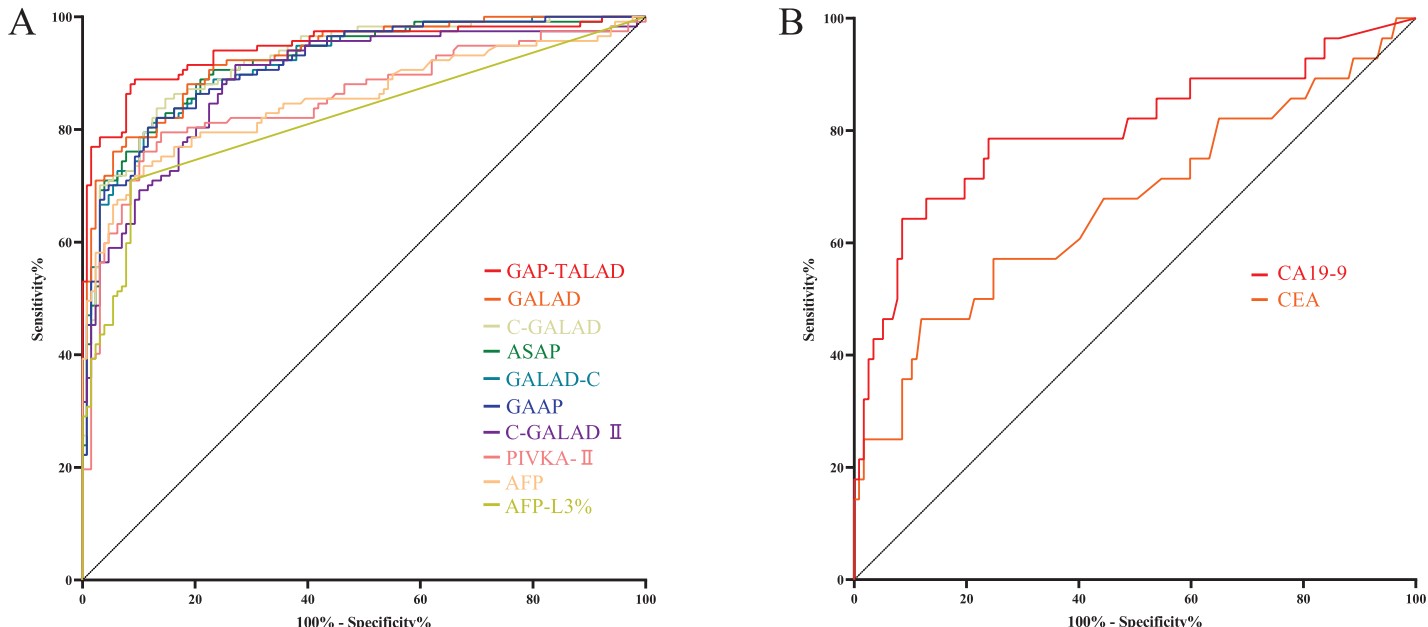

**Figure 3 ROC curves of serum tumor markers and diagnostic models in diagnosing HCC.** (A) ROC curves of GAP-TALAD, GALAD, C-GALAD, ASAP, GALAD-C, GAAP, C-GALAD II, PIVKA-II, AFP and AFP-L3% in diagnosing HCC and non-HCC (CCA and benign liver diseases). AUC was 0.9445, 0.9288, 0.9227, 0.9190, 0.9174, 0.9155, 0.8890, 0.8537, 0.8535 and 0.8197 respectively. (B) ROC curves of CA19-9 and CEA in distinguishing CCA and HCC. AUC was 0.8007 and 0.6670, respectively.

**Table 4 Efficacy of serum tumor markers and diagnostic models in diagnosing liver cancers (HCC and CCA).**

| Serum tumor marker/model | AUC (95% CI)[#] | Cut-off value | Sensitivity% | Specificity% | Youden index |
|---|---|---|---|---|---|
| GALAD-C | 0.9204 [0.8875~0.9533][a] | 0.13[♦] | 82.07% | 90.10% | 0.7217 |
| GAAP | 0.9196 [0.8865~0.9526][a] | 0.29[♦] | 73.79% | 98.02% | 0.7181 |
| GALAD | 0.9141 [0.8794~0.9488][a] | 0.68[♦] | 80.69% | 90.10% | 0.7079 |
| ASAP | 0.9133 [0.8778~0.9487][a] | −0.39[♦] | 84.83% | 87.13% | 0.7196 |
| C-GALAD | 0.9099 [0.8737~0.9462][a] | 1.30[♦] | 78.62% | 92.08% | 0.7070 |
| C-GALAD II | 0.8488 [0.8007~0.8968][b] | −1.50[♦] | 77.93% | 78.22% | 0.5615 |
| PIVKA-II | 0.8276 [0.7765~0.8788][c] | 40.00 mAU/mL[*] | 72.41% | 86.14% | 0.5855 |
| AFP-L3% | 0.7795 [0.7225~0.8366][c] | 10.00%[*] | 46.21% | 95.05% | 0.4126 |
| CEA | 0.6935 [0.6255~0.7614][d] | 5.00 ng/mL[*] | 18.62% | 91.09% | 0.0971 |
| CA19-9 | 0.6834 [0.6165~0.7503][d] | 34.00 U/mL[*] | 40.69% | 84.16% | 0.2485 |

**Notes:**
[#] Diagnosing liver cancers (HCC and CCA) and benign liver diseases, comparisons of AUC were conducted by DeLong tests.
[♦] According to the highest Youden index.
[*] According to the manufacturers' instructions.
[a] There was no significant difference ($p > 0.05$) in AUC among GALAD-C, GAAP, GALAD, ASAP and C-GALAD; AUC of GALAD-C, GAAP, GALAD, ASAP and C-GALAD was significantly higher ($p < 0.05$) than that of C-GALAD II, PIVKA-II, AFP-L3%, CEA and CA19-9.
[b] AUC of C-GALAD-II was significantly higher ($p < 0.05$) than that of AFP-L3%, CEA and CA19-9.
[c] There was no significant difference ($p > 0.05$) in AUC between PIVKA-II and AFP-L3%; AUC of PIVKA-II and AFP-L3% was significantly higher ($p < 0.05$) than that of CEA and CA19-9.
[d] There was no significant difference ($p > 0.05$) in AUC between CEA and CA19-9.
HCC, hepatocellular carcinoma; CCA, cholangiocarcinoma; AUC, area under receiver operating characteristic (ROC) curve; CI, confidence interval; PIVKA-II, protein induced by vitamin K absence or antagonist-II; AFP, alpha fetoprotein; AFP-L3%, percentage of AFP-L3 (culinaris agglutinin strong binding) to total AFP; CEA, carcinoembryonic antigen; CA19-9, carbohydrate antigen 19-9.

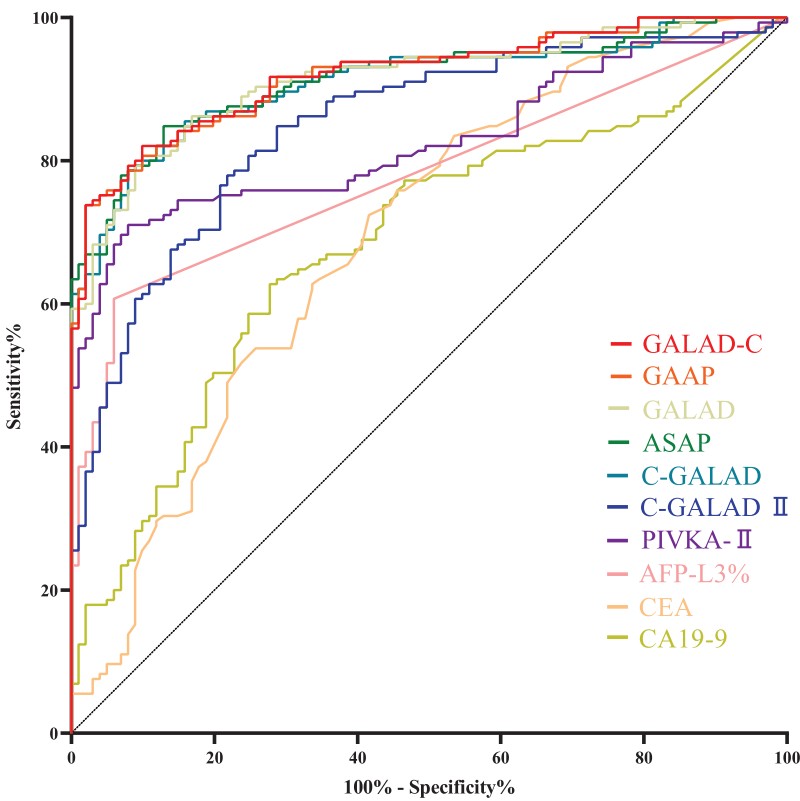

**Figure 4 ROC curves of serum tumor markers and diagnostic models in diagnosing liver cancers (HCC + CCA) and benign liver diseases.** ROC curves of GALAD-C, GAAP, GALAD, ASAP, C-GALAD, C-GALAD II, PIVKA-II, AFP-L3%, CEA and CA19-9 in diagnosing liver cancers (HCC + CCA) and benign liver diseases. AUC was 0.9204, 0.9196, 0.9141, 0.9133, 0.9099, 0.8488, 0.8276, 0.7795, 0.6935 and 0.6834, respectively.

## DISCUSSION

This study explored the optimal patterns of serum AFP, AFP-L3% and PIVKA-II for the diagnosis of liver cancers. This analysis found that the diagnostic models GALAD, ASAP, GALAD-C and C-GALAD showed the highest efficacy in the diagnosis of HCC (Table 3), liver cancers (HCC + CCA; Table 4), and early liver cancers (early HCC and CCA; Table 5). The benign liver diseases in this study included focal liver lesions that have similar clinical or imaging manifestations to liver cancers (hepatic cyst, hepatic hemangioma, hepatic abscess, focal nodular hyperplasia, angiomyolipoma, hepatic adenoma and falciform ligament pseudolesion), as well as main risk factors for the tumorigenesis of liver cancers (liver cirrhosis, chronic hepatitis and fatty liver disease), so that the results of this study could be widely applicable in the clinic.

In addition to the combined detection of serum tumor markers (AFP, AFP-L3% and PIVKA-II), the algorithms of diagnostic models can include age, gender and certain laboratory test results (TBIL, ALB and PLT). The incidence of HCC is higher in males and in individuals greater than 40 years old (*Frager & Schwartz, 2020*; *Zhou et al., 2023*). Elevated levels of TBIL can cause liver inflammation and immune deficiency, which can

**Table 5 Efficacy of serum tumor markers and diagnostic models in diagnosing early liver cancers[#].**

| Diagnosing/distinguishing | Serum tumor marker/ model | AUC (95% CI)[♦] | Cut-off value | Sensitivity % | Specificity % | Youden index |
|---|---|---|---|---|---|---|
| Early liver cancers [#] and benign liver diseases | GALAD | 0.9219 [0.8728~0.9710][a,b] | 0.68[*] | 78.38% | 90.10% | 0.6848 |
| | GALAD-C | 0.9079 [0.8537~0.9622][a,c] | 0.13[*] | 67.57% | 92.08% | 0.5965 |
| | GAAP | 0.9050 [0.8504~0.9596][a,d] | 0.29[*] | 56.76% | 98.02% | 0.5478 |
| | C-GALAD | 0.8991 (0.8399~0.9584)[a,e] | 1.30[*] | 70.27% | 92.08% | 0.6235 |
| | ASAP | 0.8946 [0.8341~0.9550][a,e] | −0.39[*] | 78.38% | 87.13% | 0.6551 |
| | C-GALAD II | 0.8293 [0.7568~0.9017][f] | −1.50[*] | 72.97% | 78.22% | 0.5119 |
| | PIVKA-II | 0.7184 [0.6109~0.8258][g] | 40.00 mAU/mL[*] | 48.65% | 86.14% | 0.3479 |
| | AFP-L3% | 0.7173 [0.6081~0.8265][g] | 10.00%[*] | 37.84% | 95.05% | 0.3289 |
| | CA19-9 | 0.6655 [0.5630~0.7680][g] | 34.00 U/mL[*] | 32.43% | 84.16% | 0.1659 |
| | CEA | 0.6409 [0.5440~0.7378][g] | 5.00 ng/mL[*] | 10.81% | 91.09% | 0.0190 |
| Early HCC and non-HCC[‡] | GAP-TALAD | 0.9540 [0.9195~0.9886][h] | 0.68[ǂ] | 93.10% | 90.70% | 0.8380 |
| | AFP | 0.8447 [0.7673~0.9221][h] | 7.00 ng/mL[*] | 68.97% | 86.05% | 0.5502 |

**Notes:**
[#] Including HCC with BCLC stages 0 to A and CCA with TNM stages 0 to II.
[♦] Comparisons of AUC were conducted by DeLong tests.
[*] For diagnosing liver cancers, calculated in Table 4.
[•] According to the manufacturers' instructions.
[‡] CCA and benign liver diseases.
[ǂ] For diagnosing HCC, calculated in Table 3.
[a] There was no significant difference ($p > 0.05$) in AUC (diagnosing early cancer livers and benign liver diseases) among GALAD, GALAD-C, GAAP, C-GALAD and ASAP.
[b] AUC (diagnosing early cancer livers and benign liver diseases) of GALAD was significantly higher ($p < 0.05$) than that of C- GALAD II, PIVKA-II, AFP-L3%, CA19-9 and CEA.
[c] AUC (diagnosing early cancer livers and benign liver diseases) of GALAD-C was significantly higher ($p < 0.05$) than that of PIVKA-II, AFP-L3%, CA19-9 and CEA.
[d] AUC (diagnosing early cancer livers and benign liver diseases) of GAAP was significantly higher ($p < 0.05$) than that of C- GALAD II, PIVKA-II, AFP-L3%, CA19-9 and CEA.
[e] AUC (diagnosing early cancer livers and benign liver diseases) of C-GALAD and ASAP was significantly higher ($p < 0.05$) than that of PIVKA-II, AFP-L3%, CA19-9 and CEA.
[f] AUC (diagnosing early cancer livers and benign liver diseases) of C-GALAD II was significantly higher ($p < 0.05$) than that of AFP-L3%, CA19-9 and CEA.
[g] There was no significant difference ($p > 0.05$) in AUC (diagnosing early cancer livers and benign liver diseases) among PIVKA-II, AFP-L3%, CA19-9 and CEA.
[h] AUC (diagnosing early HCC and non-HCC) of GAP-TALAD was significantly higher ($p < 0.05$) than that of AFP.
AUC, area under receiver operating characteristic (ROC) curve; CI, confidence interval; PIVKA-II, protein induced by vitamin K absence or antagonist-II; AFP, alpha fetoprotein; AFP-L3%, percentage of AFP-L3 (culinaris agglutinin strong binding) to total AFP; CA19-9, carbohydrate antigen 19-9; CEA, carcinoembryonic antigen; HCC, hepatocellular carcinoma; BCLC, Barcelona clinic liver cancer; CCA, cholangiocarcinoma; TNM, primary tumor, regional lymph nodes and distant metastasis.

accelerate the progression of tumors (*Xu et al., 2023*). ALB has a direct role in HCC growth inhibition, either through modulation of AFP or through its actions on growth-controlling kinases (*Bagirsakci et al., 2017*). The production of thrombopoietin (TPO) can decrease when liver functions are damaged, so the reduction of blood PLT frequently occurs in HCC patients (*Kurokawa & Ohkohchi, 2017*). Studies also found that the increase of TBIL and decrease of ALB and PLT were the main laboratory parameters (other than tumor markers) that could predict the occurrence of HCC (*Fan et al., 2020*; *Yao, Zhao & Lu, 2016*). The current study found that the efficacies of the selected diagnostic models were all higher than those of single detections of the serum tumor markers in the diagnosis of HCC, liver cancers (HCC + CCA) and early liver cancers. This indicated that demographic information and certain laboratory tests (TBIL, ALB and PLT) could improve the diagnostic efficacy for liver cancers and highlighted the significance of the combined detection of serum AFP, AFP-L3% and PIVKA-II in the diagnosis of liver cancers. All the
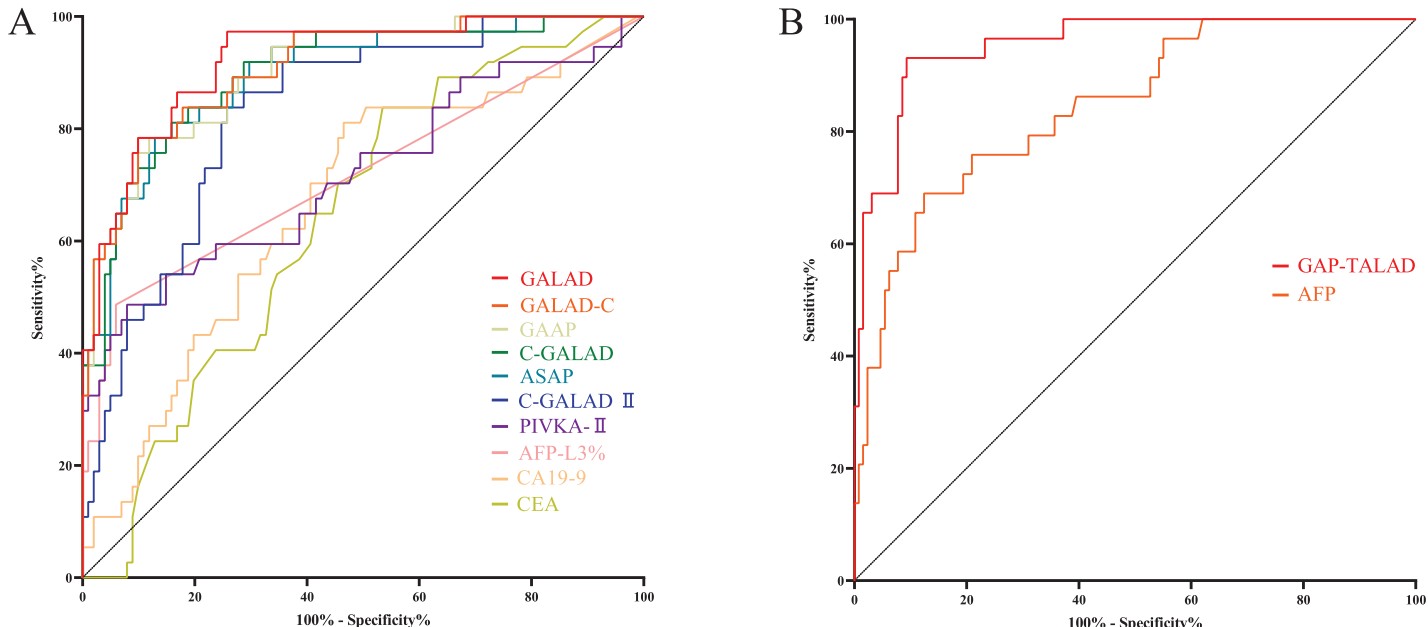

**Figure 5  ROC curves of serum tumor markers and diagnostic models in diagnosing early liver cancers.** (A) ROC curves of GALAD, GALAD-C, GAAP, C-GALAD, ASAP, C-GALAD II, PIVKA-II, AFP-L3%, CA19-9 and CEA in diagnosing early liver cancers (including early HCC and CCA) and benign liver diseases. AUC was 0.9219, 0.9079, 0.9050, 0.8991, 0.8946, 0.8293, 0.7184, 0.7173, 0.6655 and 0.6409, respectively. (B) ROC curves of GAP-TALAD and AFP in distinguishing early HCC and non-HCC (CCA and benign liver diseases). AUC was 0.9540 and 0.8447, respectively

diagnostic models had different levels in HCC and CCA. HCC develops from hepatocytes, while CCA arises from the bile duct epithelium (*Zhou et al., 2023*; *Shin, Moon & Kim, 2023*). AFP, AFP-L3% and PIVKA-II are all produced by hepatocytes and are more specific to HCC. HCC and CCA also have differences in gender composition; most HCC patients are male, but this gender disparity is not as significant among CCA patients (*Frager & Schwartz, 2020*; *Sarcognato et al., 2021*). CCA often causes obstructive changes in liver functions, and because of this, the elevation of TBIL often occurs in CCA patients (*Khan et al., 2012*; *Shin, Moon & Kim, 2023*). There are few reports on the decrease of PLT and ALB (before treatments) caused by CCA; however, the obstruction and cholestasis caused by CCA could lead to hepatocellular damage (*Shin, Moon & Kim, 2023*), which also has the possibility of influencing the levels of PLT and ALB. Based on the results of this study and the clinical characteristics of HCC and CCA, AFP, AFP-L3%, PIVKA-II and gender composition are likely the main factors leading to the different levels of the diagnostic models in HCC and CCA. This study demonstrated the effectiveness of diagnostic models of AFP, AFP-L3% and PIVKA-II detection in distinguishing HCC from CAA.

Levels of diagnostic models GALAD, ASAP, GALAD-C, GAAP, C-GALAD and C-GALAD II in CCA patients were also significantly higher than those in patients with benign liver diseases. From the parameters used in the diagnostic models (age, gender, AFP, AFP-L3%, PIVKA-II, TBIL, ALB and PLT), age and levels of PIVKA-II and AFP-L3% were significantly different between CCA patients and patients with benign liver diseases (Table 2), and this may be an explanation for the differences in levels of the

diagnostic models between CCA patients and patients with benign liver diseases. The incidence of CCA is higher in individuals over 50 years of age (*Sarcognato et al., 2021*). CCA often causes cholestasis, which can increase the production of PIVKA-II (*Tameda et al., 2013*). A subgroup of intrahepatic CCA patients have clinicopathologic features resembling those of HCC, and their levels of AFP-L3% may also be elevated (*Okuda et al., 2006*). The GAP-TALAD model did not show different levels between CCA and benign liver diseases, and this may be due to the larger weight of nonspecific indicators (TBIL, ALB and PLT) in the formula of the GAP-TALAD model (Table 1), masking the influence of ages and levels of tumor markers in CCA patients to the formula. There have been limited studies that analyzed levels of diagnostic models in CCA, and the results of this study revealed the importance of further investigations into the diagnostic models using AFP, AFP-L3% and PIVKA-II detection in CCA patients, and of assessing the potential of these diagnostic models in the overall diagnosis of liver cancers (HCC and CAA).

Benign liver diseases also often cause damage to liver functions (liver cirrhosis, hepatitis, *etc*.), which may be the reason for the relatively small differences in levels of TBIL, PLT and ALB among HCC, CCA and benign liver diseases patient cohorts (Table 2). The results of this study suggest that TBIL, PLT and ALB can effectively improve diagnostic efficacy of serum tumor markers for liver cancers but are not the main factors leading to different levels of the diagnostic models between HCC, CCA and benign liver diseases.

In the diagnosis of HCC, the GAP-TALAD, GALAD, C-GALAD, ASAP and GALAD-C models showed the highest efficacy (Table 3); in the diagnosis of overall liver cancers (HCC and CAA), the GALAD-C, GAAP, GALAD, ASAP and C-GALAD models showed the highest efficacy (Table 4); in the diagnosis of early liver cancers, the GALAD, GALAD-C, GAAP, C-GALAD and ASAP models showed the highest efficacy (Table 5). Thus, for the detection of serum AFP, AFP-L3% and PIVKA-II, the GALAD, ASAP, GALAD-C and C-GALAD diagnostic models are preferentially recommended because they showed the highest efficacy (AUC) in the diagnosis of HCC, liver cancers (HCC + CCA), and early liver cancers. Among GALAD, ASAP, GALAD-C and C-GALAD, the GALAD model had the highest accuracy (Youden index, sensitivity%+specificity%−1) in the diagnosis of early liver cancers and HCC (Tables 3 and 5), and the GALAD-C model had the highest Youden index in the diagnosis of overall liver cancers (Table 4). The ASAP model contains only two tumor markers (AFP and PIVKA-II; Table 1). Therefore, for the diagnosis of liver cancers (HCC, CCA and their early stages), based on the Youden index, the GALAD model is overall the most preferred, followed by the GALAD-C model. The ASAP model can also be chosen for accessibility or cost-effectiveness. As the models have different diagnostic sensitivity and specificity, laboratories and medical institutions can choose from GALAD, ASAP, GALAD-C and C-GALAD according to their own requirements.

The GAP-TALAD model showed the highest efficacy (AUC) in the diagnosis of HCC (Table 3), and the efficacy of GAP-TALAD in distinguishing early HCC was also higher than the efficacy of PIVKA-II, AFP-L3%, CA19-9, CEA and other diagnostic models in diagnosing early liver cancers (Table 5). These results suggested that GAP-TALAD might

be a diagnostic model that is suitable for the diagnosis of HCC (including early HCC). Therefore, the GAP-TALAD model can also be used when an HCC diagnosis is strongly suspected. However, the GAP-TALAD model cannot be used in the diagnosis of CCA (Table 2). The efficacy (AUC) of serum CEA and CA19-9 in distinguishing HCC and CCA was also lower than the efficacy of AFP, AFP-L3%, PIVKA-II and the diagnostic models in distinguishing HCC and non-HCC (Table 3). This indicated that the detection of AFP, AFP-L3% and PIVKA-II was not less valuable than CAE and CA19-9 in distinguishing CCA and HCC.

There were some limitations in this study. First, methodologies for the detection of serum tumor markers (AFP, AFP-L3% and PIVKA-II) and laboratory tests (TBIL, ALB and PLT) vary by clinical settings, so the observed efficacy of serum tumor markers and diagnostic models may be different from those in this study. Second, new diagnostic models or methods for the detection of serum AFP, AFP-L3% and PIVKA-II may also be proposed in the future, which would require the re-evaluation of the efficacy of these models. Third, results of laboratory tests TBIL, ALB and PLT can be easily affected by diseases other than liver cancers and benign liver diseases, which could impact diagnostic efficacy of the models. Fourth, as this was a retrospective study using previously collected data, the limited sample size may also have influenced the findings of this study.

## CONCLUSIONS

For serum AFP, AFP-L3% and PIVKA-II detection, the results of this study suggested that the diagnostic models GALAD, ASAP, GALAD-C and C-GALAD are recommended preferentially in the diagnosis of liver cancers (including HCC, CCA and their early stages). The diagnostic efficacy of tumor markers and diagnostic models may be influenced by varied methodologies across clinics and the presence of other diseases within patients, and new diagnostic models or methods may also be built in the future. Therefore, more studies should be conducted to further assess the efficacy of serum AFP, AFP-L3%, PIVKA-II and the diagnostic models based on these markers in different populations and methodologies, and this efficacy should also be reevaluated when new models or methods are established. The results of this study emphasized the importance of the combined detection of serum AFP, AFP-L3% and PIVKA-II in the diagnosis of liver cancers and provided clearer guidance on the selection of hematological diagnostic methods for liver cancers. Laboratories can select the patterns of serum AFP, AFP-L3% and PIVKA-II and optimize the cut-off values according to the results of this study and their own metrics.

## ACKNOWLEDGEMENTS

The authors thank all colleagues that participated in the work of serum AFP, AFP-L3%, PIVKA-II, CEA and CA19-9 detection in the Department of Clinical Laboratory, Sun Yat-Sen Memorial Hospital, Sun Yat-Sen University.

### Funding

The authors received no funding for this work.

### Competing Interests

The authors declare that they have no competing interests.

### Author Contributions

- Ling Luo conceived and designed the experiments, performed the experiments, analyzed the data, prepared figures and/or tables, authored or reviewed drafts of the article, and approved the final draft.
- Xiaochen Wang conceived and designed the experiments, performed the experiments, analyzed the data, prepared figures and/or tables, authored or reviewed drafts of the article, and approved the final draft.
- Xujian Peng conceived and designed the experiments, performed the experiments, analyzed the data, prepared figures and/or tables, authored or reviewed drafts of the article, and approved the final draft.
- Runqiang Zhong conceived and designed the experiments, performed the experiments, analyzed the data, prepared figures and/or tables, authored or reviewed drafts of the article, and approved the final draft.
- Xuejing Xuan conceived and designed the experiments, performed the experiments, analyzed the data, prepared figures and/or tables, authored or reviewed drafts of the article, and approved the final draft.
- Haixiong Lin conceived and designed the experiments, performed the experiments, analyzed the data, prepared figures and/or tables, authored or reviewed drafts of the article, and approved the final draft.
- Xianghua Lin conceived and designed the experiments, performed the experiments, analyzed the data, prepared figures and/or tables, authored or reviewed drafts of the article, and approved the final draft.
- Chaohui Duan conceived and designed the experiments, performed the experiments, analyzed the data, prepared figures and/or tables, authored or reviewed drafts of the article, and approved the final draft.

### Human Ethics

The following information was supplied relating to ethical approvals (*i.e.*, approving body and any reference numbers):

The Medical Ethics Committee of Sun Yat-Sen Memorial Hospital, Sun Yat-Sen University approved the study (SYSKY-2024-317-01).

### Data Availability

The clinical data is available in the Supplemental File.

## Supplemental Information

Supplemental information for this article can be found online at http://dx.doi.org/10.7717/peerj.19712#supplemental-information.

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
