# Peer review of "Analysis of the optimal patterns of serum alpha fetoprotein (AFP), AFP-L3% and protein induced by vitamin K absence or antagonist-II (PIVKA-II) detection in the diagnosis of liver cancers"

_PeerJ, doi:10.7717/peerj.19712_

## Round 0.1 · original submission · Major Revisions

Please address concerns of both reviewers and amend manuscript accordingly

Reviewer 1 ·

Basic reporting

This study evaluates serum biomarkers (AFP, AFP-L3%, PIVKA-a) and diagnostic models, offering a comparison across liver cancer types (HCC and CCA) and benign liver diseases.

Experimental design

The sample sizes for CCA (28 patients) and benign liver diseases (101 patients) are relatively small, which may limit the generalizability of the findings.

The comparison of diagnostic efficiency between HCC and CCA could benefit from additional focus on their distinct pathological and clinical characteristics.

While the study identifies efficient diagnostic models, it does not offer actionable recommendations for clinical implementation or prioritize models based on accessibility or cost-effectiveness.

Validity of the findings

The study may need to discuss the biological mechanisms underlying these differences, which could enhance in-depth understanding. Please refer to other studies (PMID: 37986230; PMID: 37367944) for in-depth discussion.

Additional comments

NA

Reviewer 2 ·

Basic reporting

The article by Luo L. et al. reports the findings from a study comparing diagnostic efficiency of serum AFP, AFP-L3%, PIVKA-a in liver cancers. Clinical data were collected from 117 patients with hepatocellular carcinoma (HCC), 28 patients with cholangiocarcinoma (CCA) and 101 individuals with benign liver disease. They reported that combined detection of serum AFP, AFP-L3% and PIVKA-a improved the efficiency of diagnosis while the diagnostic models GALAD, ASAP, GALAD-C and C-GALAD showed the highest efficiency in the diagnosis of HCC.
It is an interesting study that provides information that could prove to be really helpful for HCC and CCA diagnosis and could be valuable in the clinic.
However, there are a few comments, that if addressed, could greatly improve the quality of the paper:

1) The authors should carefully read the entire manuscript and make sure all grammar used is correct and all sentences are following a logical flow with connections between paragraphs and concepts. In several parts it seems like they just included a list of sentences, all of which are correct but are non interconnected.
2) Line 86-88: please edit for grammar and syntax errors. Looks like it is missing something. Same with lines 91-94, 108-110 and many other places in the text. The authors should also avoid overusing the symbol “;”. They can form smaller sentences and use full stop “.” instead.
3) Line 122: it is better to avoid saying “This study was based on the background above”. Please delete.
4) Line 130: correct “More clearer”.
5) In the methods section include formulas (instead of saying “see below”). Also, the authors may want to consider presenting the formulas in the form of a table.
6) Although I understand why informed consent was not needed, pleas include one sentence to explain it.
7) Results section needs rewriting, as it does not describe the results but rather mentions a list of findings. A more narrative format is advised.
8) The most important comment is that the authors need to provide information on the type of disease(s) diagnosed in individuals who served as the control and had “benign liver disease”. Can they be categorized in groups? Howe many per group? Is there a correlation between HCC or CCA and the type of benign disease?
9) Figure 1: age cannot be measured in levels! Please correct everywhere (in the text as well). You can say average age, age-range etc. instead.
10) Figure 1 can be split in 2 or 3 figures so that each panel becomes larger and there for legible. Right now it is very difficulty to go through the panels.

Experimental design

As mentioned in comment 8 above, the authors need to provide information on the patients with "benign liver disease" who served as controls.

Validity of the findings

No comment

---

## Round 0.2 · Minor Revisions

The tracked changes document you provided includes comments and questions from a copyeditor.

You must address these in your next revision.

Reviewer 1 ·

Basic reporting

The writing needs to be improved comprehensively.

Experimental design

N/A

Validity of the findings

N/A

Additional comments

N/A

---

## Round 0.3 · accepted · Accept

All issues were adequately addressed, and the revised manuscript is acceptable now.